# Features of Highly Homologous T-Cell Receptor Repertoire in the Immune Response to Mutations in Immunogenic Epitopes

**DOI:** 10.3390/ijms252312591

**Published:** 2024-11-23

**Authors:** Ksenia Zornikova, Dmitry Dianov, Natalia Ivanova, Vassa Davydova, Tatiana Nenasheva, Ekaterina Fefelova, Apollinariya Bogolyubova

**Affiliations:** National Medical Research Center for Hematology, Moscow 125167, Russia; kvzornikova@gmail.com (K.Z.); dvdianov@gmail.com (D.D.); halfblood394@gmail.com (N.I.); vasso4kaa@gmail.com (V.D.); dreminat@mail.ru (T.N.); katehouse15@mail.ru (E.F.)

**Keywords:** COVID-19, SARS-CoV-2, T-cell receptor, T-cell repertoire, T-cell epitope, immune evasion, peptide–MHC interaction, cross-reactivity

## Abstract

CD8+ T-cell immunity, mediated through interactions between human leukocyte antigen (HLA) and the T-cell receptor (TCR), plays a pivotal role in conferring immune memory and protection against viral infections. The emergence of SARS-CoV-2 variants presents a significant challenge to the existing population immunity. While numerous SARS-CoV-2 mutations have been associated with immune evasion from CD8+ T cells, the molecular effects of most mutations on epitope-specific TCR recognition remain largely unexplored, particularly for epitope-specific repertoires characterized by common TCRs. In this study, we investigated an HLA-A*24-restricted NYN epitope (Spike_448-456_) that elicits broad and highly homologous CD8+ T cell responses in COVID-19 patients. Eleven naturally occurring mutations in the NYN epitope, all of which retained cell surface presentation by HLA, were tested against four transgenic Jurkat reporter cell lines. Our findings demonstrate that, with the exception of L452R and the combined mutation L452Q + Y453F, these mutations have minimal impact on the avidity of recognition by NYN peptide-specific TCRs. Additionally, we observed that a similar TCR responded differently to mutant epitopes and demonstrated cross-reactivity to the unrelated VYF epitope (ORF3a_112-120_). The results contradict the idea that immune responses with limited receptor diversity are insufficient to provide protection against emerging variants.

## 1. Introduction

The adaptive immune response plays a crucial role in the successful clearance and control of SARS-CoV-2 infection. Understanding the mechanisms underlying the T-cell response to SAR-CoV-2 is important for predicting vaccine efficacy and assessing the risk of reinfection. To date, numerous studies have focused on the identification and characterization of T-cell epitopes [1,2,3,4,5], and on the SARS-CoV-2-specific T-cell receptor (TCR) repertoires [6,7,8]. More than 2200 SARS-CoV-2 antigen-specific T-cell epitopes have been found that were derived from various viral proteins, with Spike-derived ones being the most abundant [5]. Among them, several so-called immunodominant epitopes that yield strong and durable response have been found in a large portion of the population [9,10].

Robust T-cell responses to SARS-CoV-2 have been associated with mild disease and strong protection against reinfection [11,12,13]. However, the emergence of new strains has demonstrated that some mutations can lead to evasion from immune response [14]. For example, Y103D mutation in the SPRWYFYYL epitope could enhance CD8+ T-cell responses through the increased efficiency of proteasomal digestion, potentially benefiting viral fitness by enhancing CD8+ T-cell responses [15].

The P272L substitution in the Spike_269−277_ YLQPRTFLL epitope, observed in multiple viral lineages worldwide, results in the loss of a dominant HLA-A*02:01-restricted CD8+ T-cell response, leading to immune evasion. This mutation was not recognized by over 120 TCRs that responded to the founder epitope. There is an extension of the amino acid 272 side chain in the P272L variant epitope, suggesting that the longer leucine side chain might interfere with TCR binding [16]. These data were later supported by structural studies [17,18].

Mutations in the Spike_448−456_ NYNYLYRLF, such as L452R and Y453F, have also been linked to immune escape from HLA-A24-restricted cellular immunity and increased infectivity [19] without affecting HLA presentation [20].

Escape from antigen-specific CD8+ T cells has been extensively studied in other infections, such as HIV-1, where rapid intra-host evolution renders T-cell responses ineffective within weeks of acute infection [21]. Several cytotoxic T-lymphocyte (CTL) escape variants have been described in influenza, for example, R384G substitution in the HLA B*08:01-restricted NP_380−388_ and B*27:05-restricted NP_383−391_ epitopes [22]. The long-term adaptation of influenza A/H3N2 has been demonstrated, with the loss of one CTL epitope every 3 years since its emergence in 1968 [23].

Overall, a virus may evade T-cell responses through two primary mechanisms: by disrupting epitope presentation on HLA molecules or by impairing the binding of pre-existing TCRs specific to a wild-type epitope (Appendix A). Polymorphisms in HLA genes limit the advantage of escape within a particular epitope; additionally, some TCRs are able to recognize peptides presented by different HLA alleles [24,25]. Thus, the impact of viral mutations on TCR recognition should be more frequent. Indeed, it was shown that a single amino acid substitution within an epitope can suffice for complete evasion from a T-cell response, and such escape is often associated with disease progression in infections caused by rapidly mutating viruses [26,27,28]. The polyclonal T-cell response is therefore unlikely to be significantly diminished by mutations present in any circulating variant. Moreover, CD4+ and CD8+ T-cell responses in each individual recognize at least 30 to 40 SARS-CoV-2 antigen epitopes [29] and public and highly homologous TCRs comprise only about 40% of virus-specific repertoire [30]. Despite this, the emergence of only one mutation may increase infectivity and significantly decrease the immune response [19]. Following SARS-CoV-2 infection, the frequency, clonality, and diversity of epitope-specific cells decrease dramatically [31,32,33,34,35,36]. Additionally, existing memory cells may limit the expansion of naive cells during re-exposure [37]. Thus, it may significantly reduce protection from mutated epitopes, especially in the case of T-cell response with a low diversity of clonotypes. However, it should be noted that a virus can evade immune response by modifying other protein interactions [38].

In most T-cell responses, the TCR repertoires elicited by a certain antigenic epitope are diverse and distinct between individuals. In contrast, other epitope-specific TCR repertoires contain TCRs that display similarities within both an individual and across unrelated individuals, referred to as homologous TCRs. These homologous TCRs have been identified in immune responses to various human viruses including CMV, EBV [39,40], and, more recently, SARS-CoV-2 [1,40,41]. High repertoire publicity and homology are often associated with shorter CDR3 length and the increased probability of generation during VDJ recombination [42,43]. However, as virus-specific memory T-cell repertoires in circulation are shaped by antigen encounters and subsequent proliferation, public and homologous TCR sequences likely reflect highly functional T cells capable of antigen-driven proliferation. Their utility in mediating immune responses and durability against mutant variants, however, remains unclear. In this study, we used homologous TCRs specific to the NYN epitope to investigate the impact of mutations in this epitope on T-cell recognition.

## 2. Results

### 2.1. Disruption of TCR Recognition as the Primary Mechanism of Viral Immune Evasion

To investigate the relationship between the structure of the TCR repertoire and its potential reactivity with mutant homologs of immunogenic epitopes from the SARS-CoV-2 virus, we conducted an analysis of epitope-specific CDR3α and CDR3β sequences derived from VDJdb and publications [32,40,41,44,45]. Our dataset comprised 3518 unique CDR3α and 4645 unique CDR3β sequences specific to 660 MHCI-restricted epitopes from SARS-CoV-2. We refined our analysis to include only those epitopes with at least 50 associated CDR3β sequences, narrowing the scope to 22 epitopes. These epitopes predominantly originated from the ORF (*n* = 9) and S (*n* = 8) proteins, with the remaining epitopes derived from the N (*n* = 3) and M (*n* = 2) proteins. Detailed peptide descriptions including their three-letter codes are provided in Appendix A.

We assessed the level of clusterization for CDR3α and CDR3β separately, using a maximum Hamming distance of 1 (a single amino acid substitution between the two sequences, assuming they are aligned and of equal length). Most epitope-specific TCRs demonstrated diverse repertoires across both chains, with 10 epitopes showing over 20% of clonotypes forming clusters (Figure 1A). These epitope-specific repertoires were considered highly homologous. Notably, the diversity of the CDR3α and CDR3β repertoires correlated (Spearman coefficient = 0.74111, *p* = 0.00005), despite the generally lower diversity observed in the CDR3α sequences. The cluster structures varied between epitopes: for instance, epitopes such as SPR and TTD formed many small clusters, whereas epitopes like YLQ, LLY, and NYN formed 1–2 large clusters of β-chains (Figure 1B, Appendix A).

For each selected epitope, we identified mutant homologs with a single non-synonymous nucleotide substitution using data from the GISAID database (Appendix A). Mutations that occurred less than three times were excluded from further analysis. The epitope KTF exhibited the highest number of mutations (*n* = 295), while other epitopes had 40–88 mutations (Figure 1C). The average mutation frequency varied across epitopes, with NYN being the most frequently mutated (with frequency of 6.1 × 10^−6^) and SPR the least (3.1 × 10^−6^). The distribution of individual mutation frequencies also differed, with the L452R mutation in epitope NYN (0.53%) and the P822L mutation in epitope TTD (0.06%) being the most prevalent. A total of 364 mutations were localized at anchor positions; however, the mutation distribution was uniform across all positions and showed no significant correlation with TCR repertoire homology (Spearman coefficient = 0.24, *p* = 0.28) or multi-specificity (Spearman coefficient = −0.1, *p* = 0.67), which will be discussed further.

Binding strength predictions for the mutant peptides to their respective HLA molecules were conducted using NetMHCpan v4.1, focusing on the most extensively studied HLA–peptide combinations. The majority of mutations did not significantly impact the binding affinity, with only 7.3% becoming non-binders, and 82.3% and 10.4% preserving strong or weak affinity, respectively (Figure 1C). No correlation was found between binding affinity and mutation frequency (Spearman coefficient = 0.003, *p* = 0.9).

Despite the preservation of MHC-binding affinity, certain mutations were observed to evade T-cell immune responses, as demonstrated in the experimental studies (Figure 1D). We reviewed the literature that examined the T-cell immune responses to immunogenic epitopes and their mutated forms that retained MHC presentation. We were able to collect data on 12 epitopes. Most of the mutations studied (26 out of 36 unique mutations) resulted in more than a twofold reduction in immune response, while only 4 showed a less significant reduction or even an increase in response. Six mutations showed controversial results across different studies. These findings suggest that the disruption of TCR recognition, rather than MHC presentation, is the primary mechanism of viral immune evasion.

We also noted that epitopes with less diverse TCR repertoires, such as SPR, TTD, and NYN, exhibited greater resilience to mutations. In contrast, some mutations evaded the immune response of more diverse repertoires. Although this difference was not statistically significant (Figure 1E), the inclusion of additional data from this study, described in detail below, increased the statistical power to *p* = 1 × 10^−10^ (Appendix A).

The limitation of this study is the lack of mutations that had been studied using both monoclonal approaches, such as TCR cloning, and polyclonal methods, such as the stimulation of PBMCs from convalescent individuals, across all epitopes except for TTD and KTF. Specifically, the TTD mutations T819I and P822L were found to have no effect on recognition by a single TCR clone [45], but significantly reduced the polyclonal T-cell response in convalescent individuals [47]. A similar pattern was observed for KTF mutations T362I and P365S, which diminished the polyclonal T-cell response, though the T366I mutation had the opposite effect [4,47]. Thus, the response of a particular receptor may not reflect the overall response. However, we hypothesized that in the case of a highly homologous repertoire, disrupting the response of clonotypes within this dominant cluster should lead to a loss of immunogenicity for the mutated epitopes. To investigate this, we focused on the NYN epitope, characterized by a predominant clustering of CDR3β sequences within a single dominant cluster. Notably, this epitope exhibited the highest mutation frequency among those analyzed.

### 2.2. Resistance to Mutations Differs Between Homologous NYN-Specific Receptors

Among those identified within the GISAID database mutations, we selected the eight most frequent variants of the NYN epitope with a single-nucleotide substitution and three variants with dual substitutions that preserved MHC presentation as predicted by NetMHCPan (Figure 2A). Subsequently, we clustered published NYN-specific TCRs and selected four distinct TCRs (Appendix A) with CDR3β from the dominant cluster that were identified across multiple sources (Figure 2B).

To evaluate the functional properties of these selected TCRs, we transduced the Jurkat E6-1 TPR cells with lentiviral vectors encoding each TCR. The transduced cells were then stimulated using the K562 cell line, which was transgenic for HLA-A*24:02 and loaded with varying concentrations of the peptides. TCR activation was assessed by measuring eGFP expression (see Section 4 for details).

All TCRs recognized the cognate NYN epitope with functional avidities (EC50) ranging from 0.15 nM (for NYN-3) to 12.4 nM (for NYN-1). We further tested the ability of these TCRs to recognize 11 mutant variants of the NYN epitope (Table 1). Functional measurements revealed a consistent pattern of recognition across different TCRs, with some differences between cell lines. For example, NYN-1, despite showing the highest eGFP expression, had the highest EC50 to the wild-type NYN epitope (12.4 nM), with most mutations leading to an increase in functional avidity. In contrast, mutant variants induced higher eGFP expression in NYN-2 and NYN-4 compared to the wild-type NYN, although their EC50 values were similar (0.38 nM and 0.5 nM, respectively). NYN-3 had the lowest EC50 for recognizing the wild-type NYN (0.15 nM) (Figure 2C,D).

Some mutations led to a significant reduction in functional avidity: particularly for the L452R mutation, which showed a significant decrease in recognition (up to 178.6 M for NYN-1, 391.7 μM for NYN-2, 27 μM for NYN-3, and 57.8 μM for NYN-4). Similarly, the L452Q + Y453F variant exhibited a reduction in functional avidity (up to 104.7 μM for NYN-1, 0.31 μM for NYN-2, 0.41 μM for NYN-3, and 0.87 μM for NYN-4).

Furthermore, each TCR, except for NYN-1, was more susceptible to different mutations: NYN-2 showed a 114-fold reduction in the recognition of N450K and 23-fold reduction in the recognition of L455F, NYN-3 demonstrated a 426-fold reduction in the recognition of N450K, and NYN-4 demonstrated a 44-fold reduction in the recognition of Y453F (Figure 2C,D).

We also examined the impact of combined mutations including N450D + L452M, N450D + L452W, and L452Q + Y453F, found in the BA.2.3.20, BA.2.86, and C.37 strains, respectively. We found that the mutations N450D, L452M, L452W, and the N450D + L452W combination did not affect the T-cell response, while the N450D + L452M combination slightly disrupted the response of NYN-2 and NYN-3 but not NYN-1. The effects of the L452Q and Y453F mutations were TCR-dependent, but their combination led to complete evasion of the T-cell response (Figure 2D).

Unexpectedly, eight mutations increased the affinity of NYN-1, and N450D, L452W, and N450D + L452W increased the affinity of NYN-2 and NYN-3. CDR3β of these three TCRs had two amino acid differences in the same positions between each other, but different TRBV genes (TRBV2 for NYN-2 and -3 and TRBV6-1 for NYN-1), which were the first and second most common V-genes in the dominant cluster, respectively (Appendix A). 

Despite the high similarity in the CDR3β region of NYN-3 and NYN-4, the latter was less resistant to all mutations except for Y453F—the only one with a substitution polar to non-polar amino acid.

Mutations may occur not only to evade T-cell response, but also to evade neutralizing antibody response or to increase ACE binding or infectivity. To compare these effects of mutations, we analyzed publications and databases and found that mutations N450D, L452W, Y453F, L455F, and F456L, which did not affect the T-cell response, reduced recognition by at least some monoclonal neutralizing antibodies [9,48,49,50,51,52,53,54]. Combined mutations N450D + L452M, N450D + L452W, and L452Q + Y453F were also known to enhance antibody resistance [54,55,56,57].

N450D, L452W, Y453F, L455F, and F456L mutations also did not influence binding to ACE and/or infectivity [48,58,59,60,61], which may suggest the sufficiency of T-cell mediated response in the absence of neutralizing antibodies (Figure 2E). 

### 2.3. NYN-Specific TCRs Were Cross-Reactive to VYF Epitope

We found that some TCR sequences were annotated as specific to different epitopes. Such multi-specific TCRs were identified for several epitopes, with the highest percentages of β-chains observed for DTD (12%), LLY (11%), VYF (7.7%), and NYN (7.3%) (Figure 3A). The multi-specificity of TCRα chains was generally higher than that of the TCRβ chains (Figure 3A), with a similar distribution of epitope specificity (Figure 3B). However, the TCRα clonotypes showed a higher propensity for “cross-reactivity” with epitopes from different sources (Figure 3B).

In certain cases, multi-specificity was restricted to 2–3 epitopes. For instance, 16 out of 23 multi-specific NYN-specific TCRβ sequences were also identified as VYF and/or QYI-specific, whereas multi-specific YLQ-specific TCRβ sequences were also specific to a broader range of epitopes. Notably, multi-specific TCRs may be specific to epitopes presented by the same HLA allele such as NYN, QYI, and VYF (all presented by HLA-A24:02) or by different HLAs such as LLY (HLA-A*02:01) and SPR (HLA-B*07:01) (Figure 3B).

The observed multi-specificity might be genuine or could be attributed to the low sensitivity of certain detection methods. This latter hypothesis is supported by the observation that the percentage of multi-specific sequences was higher in more diverse repertoires, and for the two epitopes (NYN and LLY) that displayed both high homology and cross-reactivity, the cross-reactive TCRβ sequences were not similar to the clustered sequences (Appendix A), suggesting a randomness of CDR3 to be determined as multi-specific.

Nevertheless, we decided to further investigate NYN multi-specificity, and selected six immunogenic epitopes presented by HLA-A*24:02: VYFLQSINF, QYIKWPWYI, RYRIGNYKL, KQFDTYNLW, YYQLYSTQL, and VYIGDPAQL (Appendix A). A total of 5877 unique CDR3β sequences specific to these epitopes were identified from the Minervina et al., VDJdb, and Adaptive databases [62] (Appendix A). However, only VYF and QYI shared the same CDR3β sequences with NYN, and additionally, six sequences were shared between VYF and VYI (Appendix A). Moreover, 33 VYF and 20 QYI CDR3β sequences were similar to the sequences from the main NYN-specific cluster. However, the resemblance between VYF and QYI was higher than between any of them and NYN (Figure 3C). Despite the presence of CDR3β specific to both VYF and VYI, the remaining sequences were not similar. The RYR, KQF, and YYQ epitopes did not demonstrate homology between any of the studied epitopes.

In order to verify whether the observed multi-specificity was indeed cross-reactivity, we stimulated our NYN-specific cell lines with the K562 cell line loaded with 0.8 μM of each epitope. Although none of our TCRs were previously annotated as multi-specific, we found that all TCRs recognized VYF, but not other HLA-A*24-restricted epitopes. VYF induced the same level of GFP expression as NYN in the NYN-1 and NYN-2 cell lines, but 1.5-times lower expression in NYN-3 and NYN-4, respectively (Figure 3D).

## 3. Discussion

A large number of available TCR sequences specific to different SARS-CoV-2 epitopes and the high mutation frequency of the virus make it a good model for investigating the influence of TCR repertoire architecture on immune evasion [8]. 

We compared the similarity of 22 epitope-specific TCR repertoires obtained from different sources and demonstrated that 10 of them formed a T-cell response with homologous (more than 20% of clustered clonotypes) CDR3α and CDR3β. Previously, it was shown that the immune response to the epitopes, which we considered elicited highly homologous repertoires such as YLQ, SPR, LLY, and TTD, were more abundant in individuals who recovered from mild disease compared to individuals who recovered from severe disease [3,33,63,64]. On the other hand, T cells with more diverse repertoires such as those specific to MEV, KTF, and KCY persisted longer compared to those more homologous [32,65].

The presence of both homologous and diverse TCRs may contribute to a robust and durable antiviral immune response that prevents reinfection [66]. However, emerging new strains may challenge existing immune protection. Given the high mutation rate of SARS-CoV-2, immune escape mechanisms have been extensively investigated [67].

Our analysis revealed that only 7.3% of studied mutations caused a loss of binding to HLA, suggesting that immune escape primarily occurs through impairing the binding with pre-existing TCRs, rather than through disrupting presentation on HLA molecules.

We analyzed the experimental data on immune response to mutant variants of 12 immunodominant epitopes and demonstrated that epitope-specific T-cell lines from convalescent individuals with less diverse repertoires or particular TCRs from such repertoires were more durable against immune evasion. A limitation of this study was the lack of direct comparison between monoclonal approaches, such as TCR cloning, and polyclonal methods, such as the stimulation of PBMCs from convalescent individuals, across most epitopes. The selection of TCR for cloning may be biased and based on a particular receptor affinity and/or activity and may not represent clonotype frequency in a real immune response. Consequently, some mutations, such as TTD mutations T819I and P822L, or KTF mutations T362I and P365S, were found to have no effect on recognition by a single TCR clone [4,40], but significantly reduced the polyclonal T-cell response in convalescent individuals [50]. 

To address this limitation, we selected the NYN as an example of an epitope that elicited a highly homologous repertoire. We hypothesized that in this case, evasion from clonotypes from the main cluster would be equal to the complete evasion of mutant variance. Five selected β-chains originated from the main cluster, but had different TRBV usage and one or two amino acid differences in the CDR3 regions. Paired CDR3α were chosen based on their presence in the same repertoire. All TCRs recognized the NYN peptide with remarkably lower avidity than most of the reported thresholds, which typically range from 50 nM to 10 μM [4,17,68,69]. The pattern of recognition of mutant variants displayed some variability among cell lines. 

Among the 11 tested mutant variances, only L452R and L452Q + Y453F showed a significant decrease in recognition across all cell lines. One of the most frequent mutations, L452R, is the only RBD domain mutation that emerged in the Delta variant, but was absent in the Omicron variant [70], and has been reported to increase SARS-CoV-2 infectivity [19]. Furthermore, the L452R mutated Omicron variant enhanced infectivity by promoting cleavage of the spike protein, thereby improving its ability to infect lung tissues in humanized ACE2 mice [70].

The Y453F mutation is another dominant mutation that enhances the Spike protein’s affinity for ACE2, facilitating host adaptation [71,72], while the L452Q mutation, though less frequent, has been associated with increased effective reproduction rates of the Omicron BA.2 variant [73]. We showed that while these mutations did not significantly influence avidity on their own, their combination completely disrupted recognition.

Additionally, N450K, L455F, and Y453F caused a decrease in recognition by some of the tested cell lines. Our findings on the recognition of mutant variants were largely consistent with a previously published study by Deng et al. [20]. Recognition of N450K by TCR with TRBV6-1 (NYN-1 and both TCRs from Deng et al. [20]) remained preserved, whereas TCRs containing TRBV2 (NYN-2 and NYN-3) and TRBV6-4 (NYN-4) demonstrated reductions of up to 400-fold. Interestingly, TCR^NYN-II^ from Deng et al. was the only receptor that partially preserved the response to L452R, despite its similarity to other studied TCRs. This suggests that even highly similar receptors may provide protection against different mutant variants, indicating that evasion from one receptor does not necessarily imply evasion from a similar one. 

Unexpectedly, we found that NYN-1, which had the highest EC50 to the wild-type NYN epitope (80- to 24-fold higher than other TCRs), also demonstrated enhanced avidity for the recognition of eight mutant epitopes. This confirms a previous observation, that individual low-avidity TCRs can recognize mutant epitopes with high avidity. Low-avidity clonotypes that appear after initial infection also form a memory population, and while they may be not recruited in the immune response during the same infection, they expand in response to mutant viruses [74].

It is noteworthy that NYN-1 had the same β-chain as the TCR^NYN-I^ from Deng et al., whose response to Y453F was completely disrupted (Figure 2D). These two TCRs differed slightly in their α-chains, with one insertion/deletion and two substitutions, suggesting an important contribution of the α-chain in epitope recognition.

In addition to the similarity, some epitope-specific receptors were also found in VDJdb, as annotated to other epitopes from SARS-CoV-2 and other diseases. Most of such sequences originated from diverse epitope-specific repertoires, except for those specific to the NYN and LLY epitopes. While LLY-specific α- and β-chains were predominantly “cross-reactive” to SPR, which was presented in HLA-B*07:01, nearly half of the NYN-specific β-chains annotated as VYF- and/or QYI-specific were both presented by the same HLA-A*24:02. However, other non-multi-specific VYF- and QYI-specific receptors showed low similarity to NYN-specific receptors, and no similarity to the receptors specific to the other four immunogenic epitopes presented in HLA-A*24:02.

Interestingly, despite the lack of sequence similarity, all of the NYN-specific cell lines were activated in the presence of the VYF epitope, but not by other HLA-A*24-restricted epitopes. Furthermore, the level of activation of NYN-1 and NYN-2 in response to VYF was comparable to their activation by NYN, whereas NYN-3 and NYN-4 exhibited lower levels of activation. Previous studies have shown that IFN-γ production by VYF-specific cells was lower than that of QYI- and VYI-specific cells [33,75], probably due to the presence of low-avidity, cross-reactive NYN-specific cells.

In conclusion, despite the prominent similarity of NYN-specific β-chains, these receptors are susceptible to different naturally occurring mutations. Thus, with some limitations, we suggest that a highly homologous repertoire may be sufficient to provide effective protection against new viral variants.

## 4. Materials and Methods

### 4.1. Mutant Peptide Candidates

The GISAID database (accessed on 17 March 2024) [76] was used to search for homologous peptides with one non-synonymous nucleotide substitution. Mutations that occurred less than 3 times were excluded from the analysis. The affinity of peptide binding to HLA was evaluated using the NetMHCpan 4.1 program (accessed on 15 July 2024) [77]. Epitopes with a score below 0.5 were classified as strong binders, those with a score between 0.5 and 2 were considered as weak binders, and epitopes with a score exceeding 2 were identified as non-binders (Appendix A).

### 4.2. TCR Repertoire Analysis

TCR sequences with known antigen specificities were downloaded from the VDJdb database [78] and publications [32,40,41,44,45,46] (Appendix A). Epitope-specific TCR sequences were analyzed in R (version 4.0.0) and Python (version 3.9.1). Clusterization was performed with a maximum Hamming distance of 1. Graphs were plotted using “igraph” R package version 1.2.6.

### 4.3. Molecular Cloning and Cell Lines

V-regions corresponding to selected TCR chains were synthesized using oligonucleotide-annealing approach [79], fused with corresponding C-regions via overlap extension PCR and cloned to bicistronic lentiviral expression vector under the control of the EF1a promoter using GoldenGate assembly with BpiI endonuclease. In order to ensure equimolar expression, alpha- and beta-chains were separated with the P2A self-cleaving peptide. All molecular cloning work was conducted using the NEB Stable *E. coli* strain. The sequence of resulting plasmids was verified by the Sanger method.

Lentiviral particles were generated as described in Section 4. K562 cells (ATCC CCL-243) were previously transduced with A*24:02-bearing particles. The surface expression of the HLA I class was confirmed via flow cytometry using antibodies to HLA-A, HLA-B, and HLA-C (PE-conjugated, BioLegend, San Diego, CA, USA, clone W6/32, cat.311406). The Jurkat E6-1 triple parameter reporter (NFAT-eGFP, NF-κB-CFP and AP-1-mCherry) CD8+ TCR KO cells (Jurkat E6-1 TPR) [80] were transduced with TCR-bearing lentiviral particles, and the surface expression was confirmed by CD3 staining (AF700-conjugated, Sony, San Jose, CA, USA, clone OKT3).

Transduced cells were sorted with FACS Aria III (BD Biosciences, Franklin Lakes, NJ, USA) to a 98%+ purity. After sorting, the cells were cultivated in IMDM (Gibco, Thermo Fisher Scientific, Waltham, MA, USA) containing 10% FBS (Sartorius, Göttingen, Germany) and 1% penicillin-streptomycin (50 units/mL of penicillin and 50 μg/mL of streptomycin) (Gibco, Thermo Fisher Scientific, Waltham, MA, USA).

### 4.4. Lentiviral Particle Production

The adhesive cell line of human embryonic kidney HEK293T was used for lentiviral vector (LV) production. The expression plasmid was co-transfected into cells with three packaging plasmids: pLP1 (gag/pol, encodes the virion structural proteins as well as the enzymes revertase and integrase), pLP2 (rev, contains the sequence of the export protein of the non-spliced genomic Rev mRNA), and pLP/VSV-G (encodes the envelope protein, glycoprotein G of vesicular stomatitis virus) by using linear polyethylenimine (PEI 25K, Polysciences, Niles, IL, USA). DMEM (Gibco, Thermo Fisher Scientific, Waltham, MA, USA) containing 5% FBS (Sartorius, Göttingen, Germany) without added antibiotics and Opti-MEM (Gibco, Thermo Fisher Scientific, Waltham, MA, USA) were used as the culture media for transfection. The supernatant containing lentiviral particles was collected after 72 h and filtered through a 0.45-µm PES syringe filter (Sarstedt, Germany). The cleared supernatant was concentrated by centrifugation at 4000× *g* for 16 h at +4 °C, collected and resuspended in IMDM (Gibco, Thermo Fisher Scientific, Waltham, MA, USA), aliquoted, and frozen. The aliquots were stored at −80 °C. 

The functional titer of lentiviral preparations was assessed by the transduction of the Jurkat E6-1 TPR cell line with serial dilutions. Lentiviral transduction was performed with a ratio of infectious viral particles to cells that ensured not more than 20% transduction, thereby reducing the probability of multiple integrations.

### 4.5. Analysis of Peptide–MHC Interactions with Transgenic TCRs

All selected peptides (Table 1) used in this study (purity ≥ 95%) were synthesized by Pepmic Co. Ltd. (Suzhou, China). Each lyophilized aliquot of peptide was dissolved in dimethylsulfoxide (DMSO) with a final concentration of 40 uM and frozen in a series of aliquots. The aliquots were stored at −80 °C.

The interaction of peptides with epitope-specific TCRs expressed by the Jurkat E6-1 TPR was analyzed as follows. The Jurkat E6-1 TPR transgenic cells were incubated with antigen-presenting cells K562 expressing the HLA-A*24:02 allele for 18–20 h at 37 °C in 5% CO_2_ in a ratio of 1:2 in 200 μL of IMDM (Gibco, Thermo Fisher Scientific, , Waltham, MA, USA) containing 10% FBS (Sartorius, Göttingen, Germany) and 1% penicillin-streptomycin in a 96-well plate containing serial dilutions of each peptide (final concentrations 100; 20; 4; 0.8; 0.16; 0.032; 0.0064; 12.8 × 10^−4^; 2.56 × 10^−4^ 0.512 × 10^−4^; 0.1024 × 10^−4^; 4.096 × 10^−7^ μM) in three independent replicates. Medium containing cells without the added peptide was used as the negative control. Cells stimulated with 0.1 ug/mL of PMA (Sigma-Aldrich, St. Louis, MI, USA) and 2.5 ng/mL of ionomycin (Sigma-Aldrich, USA) were used as the positive control. An irrelevant peptide in the same serial dilutions was used in the test as a control to prevent false positive results; no inhibition was observed.

After incubation, cells were washed with PBS/BSA/EDTA solution (phosphate-buffered saline, containing 0,5% bovine serum albumin and 2 mM ethylene diamine tetraacetic acid) and stained with fluorescent CD8 antibody conjugated to APC (BD Biosciences, Franklin Lakes, NJ, USA, cat.345775). Cells were incubated for 20 min at 4 °C and then washed with PBS/BSA/EDTA solution. Jurkat E6-1 TPR cell activation was assessed by eGFP expression regulated by the NFAT promoter and analyzed using a MACSQuant Analyzer 10 (Miltenyi Biotec, Bergisch Gladbach, Germany). Flow cytometry data were analyzed using FlowJo (version.10.7.1, BD Biosciences, Ashland, OR, USA) and GraphPad Prizm (version 8.0, New York City, NY, USA) software. Percent of eGFP expression cells was calculated in the CD8+ gate. Negative control values were subtracted (Appendix A).

### 4.6. Statistical Analysis

All data analysis was performed using GraphPad Prism (version 8.0, New York City, NY, USA) software and Python (version 3.9.1). The peptides’ HLA binding affinity score and rank were predicted by NetMHCpan 4.1. Epitope-specific TCR sequences were analyzed in R (version 4.0.0) and Python (version 3.9.1).

## Figures and Tables

**Figure 1 ijms-25-12591-f001:**
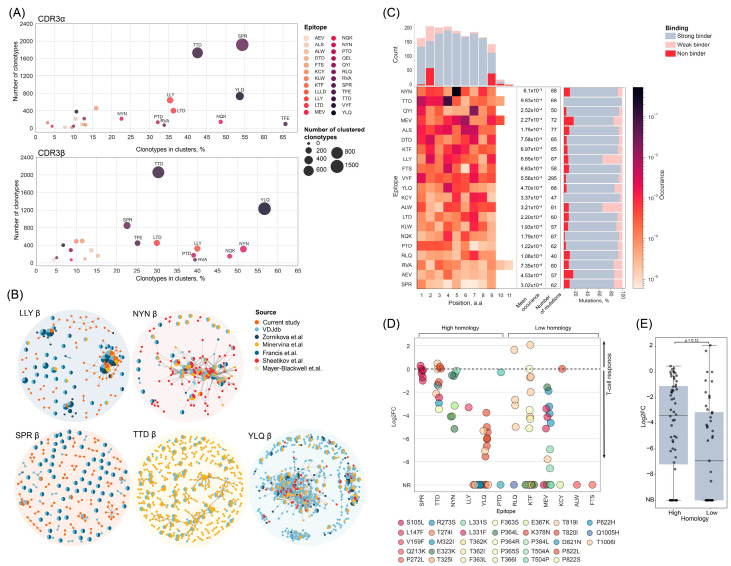
(**A**) The percentage of clustered epitope-specific CDR3α (**top**) and CDR3β (**bottom**). (**B**) Clusters of CDR3β sequences of epitope-specific CD8+ T cells. Each node represents a unique CDR3β amino-acid sequence or public sequence, the node size is proportional to the number of identical clonotypes. Lines connect similar CDR3β sequences, with Hamming distance  =  1. Colors indicate source. Only clusters with two or more members are shown [8,31,40,41,45,46]. (**C**) Occurrence of mutations in selected epitopes and number of mutations in each position that influence binding affinity to MHC predicted with NetMHCPan4.1 (**on top**). (**D**) Log2 fold change of immune response to mutated epitope compared to wild type from the experimental studies. Colors indicate different mutations. (**E**) Comparison of immune response to mutated epitopes between repertoires with low and high homology of TSRs. Each dot represents log2 fold change of immune response to mutated epitope compared to wild type. Mann–Whitney *p* value is shown. Data from the current study were not included.

**Figure 2 ijms-25-12591-f002:**
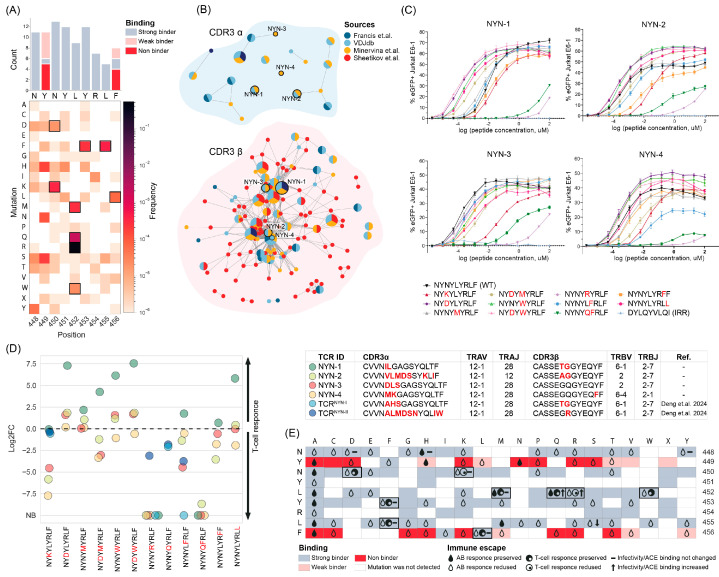
(**A**) Occurrence of mutations in the NYN epitope and number of mutations in each position that influenced the binding affinity to MHC predicted with NetMHCPan4.1 (on top). Mutations selected for this study highlighted with a frame. (**B**) Clusters of NYN-specific CDR3α and CDR3β sequences. Each node represents a unique amino-acid sequence or public sequence; the node size is proportional to the number of identical clonotypes. Lines connect similar CDR3β sequences, with Hamming distance = 1. Colors indicate source. TCRs selected for evaluation of the functional properties highlighted with a frame [31,40,41,46]. (**C**) J76 E6-1 cell line with transgenic TCR were co-cultivated with the K562-A*24:02 cell line loaded with various concentrations of NYN, mutant peptides, or irrelevant (IRR) peptides (*n*  =  3 independent replicates). T-cell activation was measured by eGFP expression regulated by the NFAT promoter. Plotted are the mean share of eGFP+ cells and SD. The studied receptor is indicated above each graph. (**D**) Log2 fold change of functional avidities (EC50) of NYN compared to mutated epitopes. Colors indicate different TCRs, which provided in the table. TCR^NYN-I^ and TCR^NYN-II^ are from Deng et.al. [20]. (**E**) Impact of mutation on T cell, antibody (AB)-mediated immune response, or virus infectivity/ACE binding.

**Figure 3 ijms-25-12591-f003:**
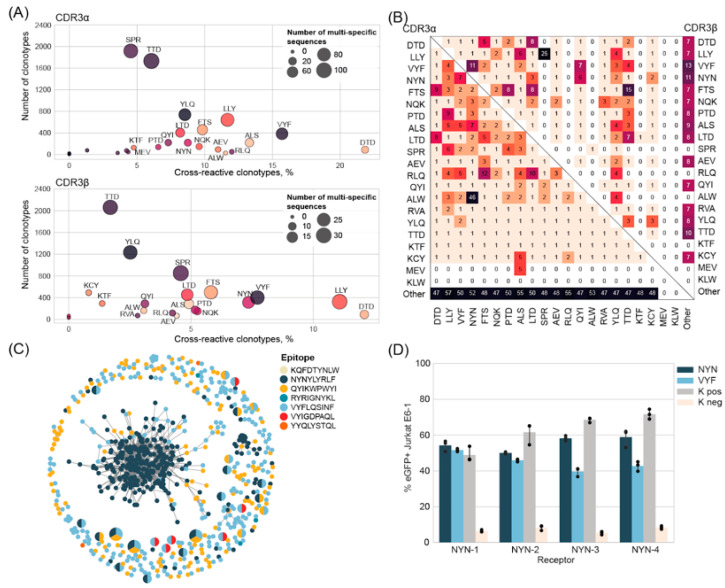
(**A**) The percentage of epitope-specific CDR3α and CDR3β that were also found to be specific to other epitopes. (**B**) Heatmap illustrates the epitope composition of multi-specific CDR3α (lower left triangle) and CDR3β (upper right triangle). (**C**) Clusters of CDR3β sequences specific to immunogenic epitopes from HLA-A*24:02. Each node represents a unique amino-acid sequence or public sequence; the node size is proportional to the number of identical clonotypes. Lines connect similar CDR3β sequences, with Hamming distance  =  1. Colors indicate epitope. Only clusters containing sequences with different specificity are shown. (**D**) The mean expression of eGFP+ of Jurkat E6-1 TPR cells stimulated with the K562 cell line loaded with 0.8 μM of NYN or VYF peptides.

**Table 1 ijms-25-12591-t001:** Strong-binding selected peptides for HLA-A*24:02.

Peptide Sequence	Type	Position
NYNYLYRLF	Wild	S448–456
NYNYLFRLF	Mutant	Y453F
NYNYLYRFF	Mutant	L455F
NYNYMYRLF	Mutant	L452M
NYNYQFRLF	Mutant	L452Q, Y453F
NYNYRYRLF	Mutant	L452R
NYNYLYRLL	Mutant	F456L
NYKYLYRLF	Mutant	N450K
NYNYWYRLF	Mutant	L452W
NYDYLYRLF	Mutant	N451D
NYDYMYRLF	Mutant	N451D, L452M
NYDYWYRLF	Mutant	N451D, L452W
DYLQYVLQI	Irrelevant	–
VYFLQSINF	Cross-reactive	–

## Data Availability

The original datasets are presented in the article and Appendix A. Further inquiries can be directed to the corresponding author.

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
