# Peer review of "Features of Highly Homologous T-Cell Receptor Repertoire in the Immune Response to Mutations in Immunogenic Epitopes"

_ijms, 2024, doi:10.3390/ijms252312591_

Round 1
Reviewer 1 Report
Comments and Suggestions for Authors
The article refers to highly homologous T cell receptor repertoire in the immune response to mutations in immunogenic epitopes to bind and present SARS-CoV-2 viral proteins/peptides. The article has a good rationale. The methodology is adequate, the Figures are explanatory, and the Table has relevant information on the different structures. The article is suitable for publication; however, I have two suggestions. The abstract must be slightly modified so the conclusion is a separate sentence. This is crucial since the conclusion has a substantial impact on the topic. The second point refers to the limitations of the model analyzing ORF3a biological function. This protein binds to Golgi and may interact with other proteins in the process and generating viral escape https://pmc.ncbi.nlm.nih.gov/articles/PMC9972675/#:~:text=4)%3A106280.%20doi%3A-,10.1016/j.isci.2023.106280,-Coronavirus%20accessory%20protein
Author Response
We thank the Reviewers for their thorough examination of the work and for their valuable comments and suggestions. The text and figures have been edited to address the issues raised by the Reviewers. A point-by-point response to the Reviewers’ comments is presented below.
Comment 1: The abstract must be slightly modified so the conclusion is a separate sentence.
Response 1: Thank you for this suggestion, we changed the abstract.
Comment 2: The second point refers to the limitations of the model analyzing ORF3a biological function. This protein binds to Golgi and may interact with other proteins in the process and generating viral escape https://pmc.ncbi.nlm.nih.gov/articles/PMC9972675/#: ~:text=4)%3A106280.%20doi%3A-10.1016/j.isci.2023.106280,-Coronavirus%20accessory%20protein.
Response 2: We added additional information regarding this limitation to emphasise that there may be other mechanisms of viral escape (lines 78 -79).
Reviewer 2 Report
Comments and Suggestions for Authors
This study performed bioinformatics analysis and in vitro study to identify SARS-CoV-2 epitopes and their mutation, and relate to the host immune response. The study showed the immune escape because of mutations impaired the binding of pre-existing TCRs, rather affecting the HLA mechanism.
The study focused on NYN epitope. There are few publications about this epitope. Therefore, the results are of interest.
Comments:
(1) Could you please show a figure with the structure of the spike protein and the coronavirus virus?
A good article is the following:
Du, L., He, Y., Zhou, Y. et al. The spike protein of SARS-CoV — a target for vaccine and therapeutic development. Nat Rev Microbiol 7, 226–236 (2009). https://doi.org/10.1038/nrmicro2090
(2) Regarding "Overall, a virus may evade T-cell responses through two primary mechanisms: by disrupting epitope presentation on HLA molecules or by impairing the binding of pre-existing TCRs specific to a wild-type epitope."
Could you please make a figure?
(3) What parts of the SARS-CoV-2 elicit immune response? Why the focus is on the spike area and not in other parts of the virus no subjected to such a mutational rate of change?
(4) Line 408. Could you please add the catalog numbers of the primary antibodies?
(5) Line 406. Why using a chronic myelogenous leukemia cell line?
(6) Sorry to ask, but what is the meaning of "Hamming distance = 1"?
(7) The NYN epitope (spike 448-456) creates strong T-cell response. Apart of CTL, is there any different mechanism involved? Additional innate immunity?
(8) How a highly homologous repertoire can be achieved?
(9) Should figure 1D show p values?
Author Response
We thank the Reviewers for their thorough examination of the work and for their valuable comments and suggestions. The text and figures have been edited to address the issues raised by the Reviewers. A point-by-point response to the Reviewers’ comments is presented below.
Comment 1: Could you please show a figure with the structure of the spike protein and the coronavirus virus?
Response 1: We added Supplementary figure 1 with the Spike protein structure.
Comment 2: Regarding "Overall, a virus may evade T-cell responses through two primary mechanisms: by disrupting epitope presentation on HLA molecules or by impairing the binding of pre-existing TCRs specific to a wild-type epitope." Could you please make a figure?
Response 2: Thank you for the suggestion, we added the schematic representation of these mechanisms on Supplementary figure 1.
Comment 3: What parts of the SARS-CoV-2 elicit immune response? Why the focus is on the spike area and not in other parts of the virus no subjected to such a mutational rate of change?
Response 3: All parts of SARS-CoV-2 are immunogenic; however, our choice of the model epitope (NYN) was driven by the limited availability of T-cell receptor sequences rather than the epitope's origin. Among the 660 epitopes analyzed, only 22 had at least 50 associated CDR3β sequences. Moreover, the NYN epitope elicited the least diverse repertoire among these 22 epitopes.
Comment 4: 408. Could you please add the catalog numbers of the antibodies?
Response 4: The catalog numbers were added.
Comment 5: Line 406. Why using a chronic myelogenous leukemia cell line?
Response 5: Jurkat E6-1 triple parameter reporter (NFAT-eGFP, NF-κB-CFP and AP-1-mCherry) CD8+ TCR KO cells (Jurkat E6-1 TPR) was specifically generated to study T cell responses to tumor and virus antigens (10.18632/oncotarget.24807). It provides more consistent results compared to primary T cells derived from convalescent donors. The latter may be influenced by donor-to-donor variability.
Comment 6: Sorry to ask, but what is the meaning of "Hamming distance = 1"?
Response 6: Hamming distance = 1 implies a single amino acid substitution between the two sequences, assuming they are aligned and of equal length. We added the description (Line 106) as well.
Comment 7: The NYN epitope (spike 448-456 ) creates strong T-cell response. Apart of CTL, is there any different mechanism involved? Additional innate immunity?
Response 7: While overall innate immunity is activated in response to SARS-CoV-2 and the virus employs various mechanisms to evade innate defense systems, there is no evidence that the NYN epitope or any Spike-derived peptides have an impact on these processes.
Comment 8: How a highly homologous repertoire can be achieved?
Response 8: Since the structure of a repertoire is determined by the nature of the epitope, it cannot be artificially designed. However, our study suggests that T-cell receptors derived from homologous repertoires, or the epitopes that elicit them, could be utilized in vaccines or TCR-based therapies.
Comment 9: Should figure 1D show p values?
Response 9: Thank you for the comment. However, in Figure 1D, our aim was to show the trend rather than to statistically compare the Log2 Fold Change of immune responses to different mutated epitopes. This is because, for some epitopes, only a single mutation has been described (e.g., mutation S105L in the SPR epitope), while for others, the effects of multiple mutations were shown (e.g., 3 mutations in YLQ and 5 mutations in MEV). We conducted a statistical comparison of these values, but grouped them by repertoire type, as shown in Figure 1E.